# Assessing the Impact of COVID-19 Phased Vaccine Eligibility on COVID-19 Vaccine Intent among African Americans in Southeastern Louisiana: A Community-Based, Cohort Study

**DOI:** 10.3390/ijerph192416737

**Published:** 2022-12-13

**Authors:** Sara Al-Dahir, Martha Earls, Christopher Gillard, Brittany Singleton, Erica Hall

**Affiliations:** College of Pharmacy, Xavier University of Louisiana, New Orleans, LA 70125, USA

**Keywords:** vaccine intent, vaccine hesitancy, COVID-19 vaccine, vaccine eligibility, African American

## Abstract

The purpose of this study was to explore the impact of eligibility for the coronavirus 2019 (COVID-19) vaccine at the time of the vaccine rollout as a predictor of vaccine intent within the African American community. Methods: Four hundred eighty-seven African American participants in southeastern Louisiana were surveyed from January–April of 2021, with follow-up surveys occurring in Fall 2021. Survey domains included demographics, vaccine hesitancy, discrimination in the healthcare setting, and knowledge and experiences with COVID-19. Descriptive statistics, Chi-square tests, and binary logistic regression were performed. Results: Participants eligible for the vaccine were 1.61 times as likely to express positive vaccine intent versus ineligible participants. Additional predictors of vaccine intent were age, insurance status and coverage, and female sex at birth. In the multivariable logistic analysis, eligible individuals were 2.07 times as likely to receive the vaccine versus ineligible individuals. Conclusions: Vaccine eligibility for the COVID-19 vaccine was a significant predictor of intent to vaccinate in the African American community. Younger individuals were less likely to have a positive intent, correlating with the eligibility of ages 16+ occurring 5 months post-vaccine approval.

## 1. Introduction

Despite federal approval of several coronavirus 2019 (COVID-19) vaccines and the consequent vaccination of millions of people worldwide, the COVID-19 pandemic continues to pose a global threat. In order to combat the virus, widespread vaccination across diverse population groups is required. Researchers discovered that African Americans, who were disproportionately affected by COVID-19, had lower vaccination intentions at the start of the pandemic [1,2,3]. There are several plausible explanations for vaccine hesitancy. African Americans, in comparison to other respondents, had a more favorable attitude toward the mainstream media and public health officials; however, some African Americans disregard information/directives from government and public health experts, due to generational distrust, prior racial discrimination, and poor communication [1,4]. Additionally, published studies identified the virus’s politicization and media messaging as a new impediment to severe acute respiratory syndrome coronavirus 2 (SARS-CoV-2) vaccine reluctance [5,6,7,8].

African Americans’ vaccine hesitancy is not limited to COVID-19; a 2012 National Health Interview Survey of adult vaccines (influenza, tetanus, pneumococcal, and HPV) revealed significantly lower vaccination rates for African Americans, despite the fact that the barrier profile was similar to COVID-19 [9,10]. Numerous studies on COVID-19 vaccination hesitancy reveal that sociodemographic and cognitive factors are significant predictors of COVID-19 vaccine hesitancy in minorities [11,12,13]. These barriers to vaccination hesitancy are not unique to COVID-19; numerous roadblocks to healthcare existed prior to the pandemic (i.e., younger individuals, females, lower income, lower education, and low literacy) across racial/ethnic groups [14,15]. Despite these long-standing barriers to healthcare, African American workers are overrepresented in many front-line jobs, and those with low wages are more likely to be exposed to COVID-19; therefore, vaccination should be a priority for this population [11]. We conducted a survey of a diverse and unique population to ascertain differences in vaccine intent and significant concerns about obtaining the COVID-19 vaccine. While Black/African American and American Indian/Alaska Native participants were less likely to receive the vaccine than White participants (32% and 29%, respectively), other minorities expressed a greater willingness to receive the vaccination. Overall, respondents to the survey indicated that the most significant concerns about vaccination were side effects (52%), safety (45%), and effectiveness (34%) [16].

On 22 March 2020, in New Orleans, Louisiana, the severe acute respiratory syndrome coronavirus 2 (SARS-CoV2) made its first appearance. Just a few weeks later, by late April 2020, New Orleans had established itself as the Southern United States’ COVID-19 epicenter [17]. The virus has disproportionately harmed the African American community in New Orleans and throughout the United States since its emergence. By April 2020, the prevalence of COVID-19 among African Americans would be three times that of the prevalence among Whites [18]. According to the Centers for Disease Control (CDC), non-Hispanic Blacks have a 2.4-fold higher rate of age-adjusted hospitalizations for the virus than non-Hispanic Whites, equating to an incidence of 1586.7/100,000 population [19]. As of August 28, 2020, the incidence of COVID-19 was 2841/100,000 among African Americans in Orleans Parish, which was 8.7 times the national average. African Americans made up 47.6% of the 4741 fatalities in the state [17].

Using a community-based participatory research approach, this study focuses on assessing potential vaccine disparities and predictors of these disparities among communities in Louisiana. The purpose of this study is to determine the impact of vaccine eligibility on the likelihood of vaccine uptake among African Americans in southeastern Louisiana.

## 2. Materials and Methods

This was a prospective, cohort study among African Americans in Southern Louisiana. Surveys were conducted from January–April of 2021 and were completed by 487 African American participants, with follow-up surveys occurring in Fall 2021. A multicenter, community-based, participatory research approach was employed with community partners at different sites: clinics, faith organizations, community organizations, and pharmacies. A total of 433 participants across 10 community partners was needed to determine a 15% difference in vaccine uptake based upon vaccine eligibility to reach 80% power with a two-tailed alpha of 0.05, assuming a 20% loss to follow-up. This study recruited 487 individuals with only a 15% loss to follow-up. Some of the loss to follow-up was attributable to Hurricane Ida, which occurred in Fall 2021.

All researchers participated in protocol training. Recruitment of sites was chosen based upon geographic location, particularly clinics, pharmacies and faith and social centers that were identified by social vulnerability indices such as zip codes in which the mean household income was less than the state average or the social vulnerability score was >0.5, indicating moderate to high social vulnerability. Clinics and pharmacies identified as serving underserved populations were preferentially recruited.

Recruitment techniques were adaptive between in-person and randomized via provided call lists from community partners, based upon the phased lockdown and contact precautions required by the regional authorities or the institution. Civic/social organizations were largely closed but faith-based organizations (FBOs) continued. We conducted adaptive sampling to recruit more men from the barbershops based on African American socialize and universities and alumni organizations. As the vaccine became available in Louisiana in December 2020, the nine-month follow-up period through Fall 2021 was sufficient time to ensure that all participants would be eligible for the vaccine by the time of the follow-up period.

Survey domains included baseline demographics such as age, rage, language, education, and family income. In addition, social determinants of health around structural barriers such as COVID-19 testing, and vaccine access were also assessed. Several domains addressed the experience of the participant around discrimination in the healthcare setting. The impact of COVID-19 as well knowledge and experiences with COVID-19 with regard to its impact on intent to receive the COVID-19 vaccine were collected. Finally, questions around vaccine hesitancy, the COVID-19 vaccine and general vaccine uptake were also assessed.

### 2.1. Community Engagement

Our institution, Xavier University of Louisiana, has a continuing relationship via its partnership with the Louisiana Clinical and Translational Science Center, where Xavier serves as the Lead in the Community Outreach and Engagement Core. Through its trusted relationship with the community, the College of Pharmacy, the Center for Minority Health and Health Disparities Research and Education, and the Department of Public Health Sciences have been addressing the health needs of Louisiana citizens, particularly those within medically underserved communities, for many years. The College of Pharmacy has been actively involved with community-based organizations, faith-based organizations, and local neighborhood associations to host health fairs and various health promotion activities. Since March 2020, the Center for Minority Health and Health Disparities Research and Education has hosted several COVID-19 events, educating citizens about the importance of prevention and testing. These COVID-19 activities have addressed social determinants of health, and the social inequities that continue to maintain or widen the health disparities gap. The College of Pharmacy, in collaboration with the City of New Orleans, conducted COVID-19 testing within several vulnerable communities. Additionally, the College of Pharmacy has extensive core partnerships with community pharmacies and healthcare clinics that serve the southeastern Louisiana region. These institutional partnerships were instrumental in the community engagement strategies for this project.

The goal for community engagement was to develop an adequate, representative sample of the community to assess COVID-19 phased vaccine eligibility and vaccine intent with respect to education level, age, sexual identity and employment. Figure 1 demonstrates the types of community recruitment targets for this project. Faith-based organizations are important centers for health promotion and community engagement, especially in underserved communities [20]. African Americans members of several local church congregations and mosques were recruited and surveyed with the assistance of their respective organization’s faith-based leaders.

In this study, we surveyed community members at local community pharmacies, which are ideal healthcare institution partners because of the shared mission of serving the medically underserved or vulnerable populations. A 2020 study mentioned that at least 90% of the United States population’s residents of vulnerable communities live within a 5-mile radius of a community pharmacy, and that patients are 12 times more likely to visit a community pharmacist than visit a primary care provider [21]. This underscores the fact that healthcare institutions such as community pharmacies are more accessible to patients than primary care centers. Due to their strategic locations, these pharmacies address the structural barriers in healthcare access by offering extended hours, establishing necessary community trust, and being geographically close to residents, thereby decreasing the need for transportation. Our institution has several institutional partnerships with primary care clinics that serve underserved patient populations; we were able to survey participants at three of these healthcare centers in the greater New Orleans area. One of the clinics was selected as a recruitment site because it services a significant number of lesbian, gay, bisexual, transgender, queer/questioning, intersex, and asexual/agender (LGBTQIA) population in the area.

There is some gap in the literature about vaccine intent and behaviors among college students; therefore, students were recruited and surveyed at a local historically black college. African American men are often underrepresented in health-based research, particularly with regard to vaccination acceptance. Early observations of data for this project indicated that survey participants were disproportionately female. Therefore, we deliberately tried to recruit more African American men, which led to community partnerships with several barbershops in the New Orleans area that service predominantly African American male clientele.

Community members that completed the survey did receive an incentive in the form of a gift card. After data collection was completed in this cohort study, the results for each site were shared with our community partners in the form of a presentation and a COVID-19 vaccination informational session was hosted for the community members that were present. Community site partners received a monetary gift and will receive additional commemorative recognition acknowledging their community service.

### 2.2. Race

Race and ethnicity were assessed, per self-identified race of the participant. Recruitment was limited to individuals who identified as African American or black as their primary racial designation. Additional racial variables were collected as secondary identifiers. Race was included in the assessment since the purpose of the Rapid Acceleration of Diagnostics-Underserved Populations (RADx-UP) consortium was to provide education, intervention, and targeted communication to communities identified as vulnerable in the COVID-19 pandemic. African American and black individuals were identified as bearing a disproportionate burden of the COVID-19 pandemic and were identified in the pandemic’s early stages as a population less likely to receive the COVID-19 vaccine [18,22].

### 2.3. Definition of Variables

The primary outcome variable is the likelihood of vaccine uptake. In Louisiana, people had vaccine cards and verified vaccination status via mobile apps at the time of the survey. Thus, participants were able to confirm vaccination status with date and location at the time of interview. Positive vaccine uptake was determined by a cumulative analysis of variables, specifically: (1) Did the participant receive the COVID-19 vaccine? (2) Did the participant sign-up to receive the vaccine? or (3) Did the participant answer “Likely” or “Very Likely” to receive the vaccine when asked about likelihood of receiving the COVID-19 vaccine? These variables were exported to a binary outcome of positive vaccine intent vs. negative vaccine intent.

The primary predictor variable was vaccine eligibility at the time of the survey, which was defined per the phased rollout from December 2020 until May 2021. The Louisiana phased rollout is presented in Figure 2 [23]. Vaccine eligibility was scaled based upon a combination of factors associated with COVID-19 infection risk and vaccine allotment via national procurement and assignment strategies. Individuals were determined to be vaccine eligible or ineligible at the time of survey administration.

Additional Likert scale variables (1-Strongly Disagree to 5-Strongly Agree) were also converted to binary variable to demonstrate vaccine hesitancy or the extent to which the individual assessed the government-phased approach was appropriate, fair, and equitable. In particular, negative vaccine hesitancy, or a negative assessment of government strategies, were assigned a binary score of “1”, representing the individual disagreed or strongly disagreed with Likert scales around government pandemic response and vaccine accessibility. Vaccine hesitancy variables were derived from the standard World Health Organization Vaccine Hesitancy Scale. COVID-risk and other questions were standardized under the common data elements provided by the National Institute of Health (NIH) RADX-Up Consortium. COVID-19 infection self-reported risk, also a 5-point Likert scale, was converted to a binary variable. Individuals who chose “no risk” or “low risk” were assigned to decreased self-perceived risk.

### 2.4. Statistical Analysis

Descriptive statistics were collected and bivariate analysis using Chi-square and ordinal logistic regression were employed to determine baseline differences based upon vaccine eligibility at the time of the survey and are presented in Table 1. A stratified analysis was reviewed to determine if stratification based upon place of recruitment was needed. No strata-specific differences were noticed across recruitment sites, so grouped analysis is presented in the results. All variables were assessed for collinearity and interaction. The vaccine hesitancy scale was assessed for reliability and resulted in a Cronbach alpha = 0.853. Bivariate results using outcome of prediction for COVID-19 vaccine uptake are presented in Table 2. A multivariable logistic regression with Akaike information criterion determination for best fit was employed to evaluate the final prediction model for variables positively associated with likelihood of vaccine uptake in the population. All statistics were conducted in Stata IC, version 15.

## 3. Results

Differences based upon vaccine eligibility at the time of the survey are presented in Table 1. Since this study was conducted from January–April 2021, we expected age at baseline to differ between the eligible groups. Age did not remain a significant predictor in the outcomes analysis. Additional notable differences in participants based upon vaccine eligibility were sex at birth and healthcare use patterns. This study involved adaptive recruiting, which retained random sampling at the sites to compensate for a phased vaccine rollout. COVID-19 experiences were also different at baseline, with ineligible participants having fewer positive test results, though an equal proportion were tested for COVID-19. Ineligible participants also had a more negative perception of the government’s handling of the pandemic, including knowledge of vaccine accessibility, pandemic response, and race-based vaccine equity concerns.

The complete bivariate analysis of positive associations, even if excluded from the final model, is presented to illustrate important trends in health equity and COVID-19 impact concerns (Table 2). Eligibility at the time of the survey was among the strongest individual predictors of positive COVID-19 vaccine intent. Though not included in the final model, a strong predictor of COVID-19 vaccine intent was having a family member who had received the vaccine. Additional common positive associations are noted, as in other studies, such as age and sex at birth (female). COVID-19 impact concerns around delaying medical care also inclined respondents to answer positively to COVID-19 vaccine intent. Negative assessments of the government’s pandemic response or negative perceptions of race-based vaccine equity were also associated with negative COVID-19 vaccine intent.

The final predictor variables for positive COVID-19 vaccine intent included a total of eight variables with only five retaining their significance in the multivariable analysis and presented in Table 3. Though several variables did not retain their significance in the final analysis, no qualitative changes in association were noted. The primary predictor variable, which was vaccine eligibility at the time of the survey, increased in overall association, with individuals who were eligible at the time of the survey being two times as likely to receive the vaccine compared to participants who did not yet meet phased rollout eligibility (95% confidence interval 1.28, 3.35). Proxy markers for vaccine trust and healthcare utilization, specifically previous vaccine refusal, continued to retain significance with individuals who had a history of other vaccine refusal; specifically, those with a history of vaccine refusal were 0.18 times as likely to receive the COVID-19 vaccine. Those who relied on emergency rooms, urgent care, and hospitals were less likely to vaccinate when compared to individuals who used primary care physicians. Though no collinearity was noted between these variables, when taken together, they demonstrate the correlation between the retention in care in a primary care model that emphasizes preventative medicine leading to positive predictive behavior of future preventative medicine efforts (i.e., the COVID-19 vaccine). Finally, personal COVID-19 health risk assessment was inversely associated with positive COVID-19 vaccine intent. No difference between vaccine eligible and ineligible participants were noted based upon previous vaccine refusal, healthcare use, COVID-19 testing, self-perceived risk, essential worker status, and the impact of COVID-19 on healthcare access, as indicated by *p*-values > 0.05.

## 4. Discussion

This study explored the impact of eligibility for the COVID-19 vaccine at the time of the vaccine rollout as a predictor of vaccine intent within the African American community. COVID-19 vaccine eligibility at the time of the survey was a significant predictor of positive intent to vaccinate in the African American community. Participants were less likely to consider vaccination since the majority of the community was ineligible to receive the vaccine until a later phase of the vaccine rollout. Younger patients were less likely to carry a positive intent to vaccinate, which correlated with the eligibility of healthy adolescents ages 16+ occurring five months post-vaccine approval; this may be explained by the Health Belief Model, which purports that an individual’s health-related behaviors are influenced by several factors, including their own personal characteristics, the perceived threat of a disease based on an individual’s judgement of susceptibility and severity, and benefits and barriers associated with an action, external cues to action, and feelings of self-efficacy [24,25].

In addition to the phased COVID-19 vaccine rollout, information distributed by the government and media also focused on vulnerable populations such as the elderly, those with chronic disease, and healthcare workers [26]. Self-identified risk for COVID-19 was associated with likelihood of vaccine uptake. Yet, self-identified risk stratification as determined by categories provided by the US government may be associated with the general neutral to lack of trust in the US government trends noted in the survey responses. Thus, vaccine recommendations based upon US government recommendations were not resonating with all of the target populations, particularly those with a history of tenuous relationships with the US government. This type of information distribution likely influenced those outside of these vulnerable groups to perceive less susceptibility to COVID-19 infection and anticipate less disease severity if infected, which translated to a decreased likelihood of vaccination. While the staggered COVID-19 vaccine rollout was necessary to prioritize vaccination of the most at risk and vulnerable individuals, it also sent the inadvertent message that those who were not within early priority groups did not necessarily need the vaccine or could at least delay their vaccination. We also found that previous vaccine refusal was associated with negative COVID-19 vaccine intent, which was an expected outcome since patients who refuse routine vaccinations (i.e., influenza) would likely not want to receive a newly approved emergency use-authorized vaccine. African Americans trend towards being under-vaccinated with routine vaccinations, and similarly, they were less likely to want the COVID-19 vaccine [27].

Our study also found that individuals seeking routine medical care in acute care facilities were less likely to get the COVID-19 vaccine compared to those in established primary care settings. Historically, primary care physicians have had a significant role in the delivery of vaccinations to patients [28]. Preventative care through vaccination is more likely to be discussed and emphasized in primary healthcare clinics where individuals may have more of an established relationships with healthcare providers. Furthermore, one of the biggest factors that continues to influence an individual decision to seek vaccination is healthcare provider recommendation [29]. During the COVID-19 vaccine rollout, healthcare providers were typically first priority to receive the vaccine; consequently, this may have positively impacted providers’ confidence and ability to discuss vaccine benefits, as well as address concerns with their patients [30].

Younger individuals were less likely to have a positive intent to receive the COVID-19, which correlated with their delayed eligibility. This finding is likely related to low perceptions of risk associated with COVID-19, as observed in other studies [31]. Such patterns may indicate there is a need to educate younger individuals about COVID-19, the associated health risks, and the benefits of vaccination.

Our study is unique in that we specifically evaluate the impact COVID-19 vaccine eligibility had on vaccine intent within the African American community. Vaccine hesitancy is usually at its highest when a new vaccine is available; the timeline of our study dovetails with peak vaccine hesitancy because it began less than a month after emergency use authorization of the COVID-19 vaccine.

### Limitations

Our study had several limitations. First, our population sample was from the greater New Orleans area; therefore, our results may not reflect the vaccination patterns of the entire southeastern region, or state of Louisiana. The greater New Orleans area has proven to be the most COVID-19 vaccine receptive community in the region, as Orleans Parish leads the state in per parish vaccination rates [32]. Nevertheless, the state of Louisiana is one of four states with the least percentage of fully vaccinated individuals [33]. Second, we recruited patients from local pharmacies and healthcare clinics. While these businesses were valued community partners within our network, during the COVID-19 vaccine rollout, these healthcare settings quickly became vaccination sites. Since community members were actively seeking COVID-19 vaccinations at these locations, our sample may have been skewed towards vaccine intenders. Nevertheless, this potential skewing was balanced by including community partners within the social, faith-based, and cosmetology arenas. Third, we collected self-reported vaccine completion and intent, which leaves room for confounding and may be limited by an individual’s memory/recall. Often this was mitigated by the presence of vaccine cards for confirmation of vaccination status and date of administration. Fourth, there were some limitations in the study design. Whereas the study was powered to detect a 15% difference in vaccine uptake based upon eligibility criteria, a larger sample size would add to the external validity of the results. The study was designed as a longitudinal cohort study embedded within a community-based participatory research model. Thus, adaptability and input to the needs of the community partners was weighted in our outreach and recruitment methods. Additionally, prior studies regarding intent to receive COVID-19 vaccines demonstrated that patient’s intentions do not always translate into action [34]. Finally, perceptions and behaviors can shift quickly; therefore, these results may not reflect current COVID-19 vaccine attitudes [12]. Such changing attitudes may be due to the rapidly changing nature of the COVID-19 pandemic, as well as the associated response.

## 5. Conclusions

We assessed vaccine disparities and predictors of these disparities among African Americans in Southeastern Louisiana. Specifically, we examined the impact of vaccine eligibility on the likelihood of vaccine uptake. We found that COVID-19 vaccine eligibility at the time of our survey was a significant predictor of positive intent to vaccinate in the African American community. As COVID-19 vaccine distribution continues, new variants will challenge the vaccines. Community perspectives may change as data comes out regarding vaccine effectiveness against the new COVID-19 variants. The unvaccinated may be dissuaded from vaccination by the fact that many vaccinated people still contract COVID-19. Those resistant to vaccination may consider it unnecessary; however, patients should be reminded that the currently available vaccines still maintain strong protection against hospitalization and death.

The phased vaccine rollout, which was dictated by strategized vaccine procurement, distribution, and availability, led to both positive and negative vaccine-seeking behaviors among the American public. Where ethical conversations centered around “jumping the line” for vaccines, studies did not explore the impact of de-emphasizing morbidity and mortality risk among late phase eligible individuals and how that might impact vaccine intent. As infection, illness, and death risk led to prioritized vaccine eligibility based on age and comorbidity, younger and healthier African Americans who did not rely on the trusted relationships of primary care physicians were interpreting the phased rollout as being part of a cohort with decreased risk; therefore, these individuals were less likely to receive the vaccine. This finding leads to an important public health gap that must be addressed and perhaps communicated in alternate healthcare venues, in order to accurately address COVID-19 risk, vaccine hesitancy, and vaccine resistance. Among racial/ethnic minorities such as young African Americans, perceptions surrounding vaccine eligibility and self-identified risk should be further explored to improve COVID-19 vaccination.

At this time, there are no published studies on the impact of COVID-19 vaccine phased roll out on vaccine intention. Several reports center around trusted sources of information, known as trusted messengers. Yet, no studies look at the temporal relationship of messaging on vaccine intent. The public health implications for identifying appropriate messaging must include both the source of the message and the timing of the message. Risk communication, for the individual and for the community, must be broadly defined with clearly communicated strategies that are both immediate and long-term. Individual vaccine eligibility communications need to be embedded in language focusing on community vaccine strategies to promote a herd protective effective, which was done later in the vaccine communication.

## Figures and Tables

**Figure 1 ijerph-19-16737-f001:**
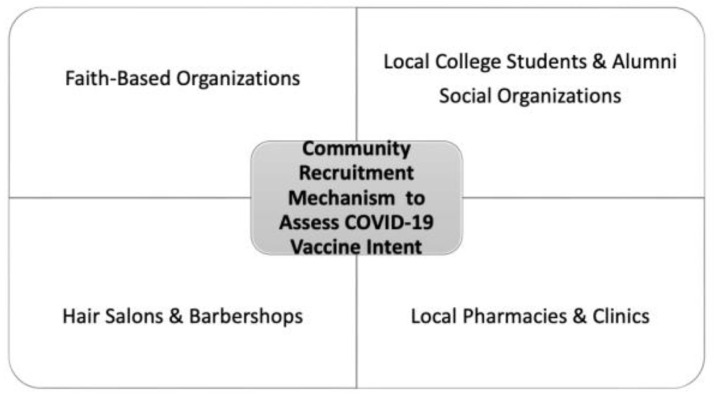
Community Recruitment Mechanism.

**Figure 2 ijerph-19-16737-f002:**
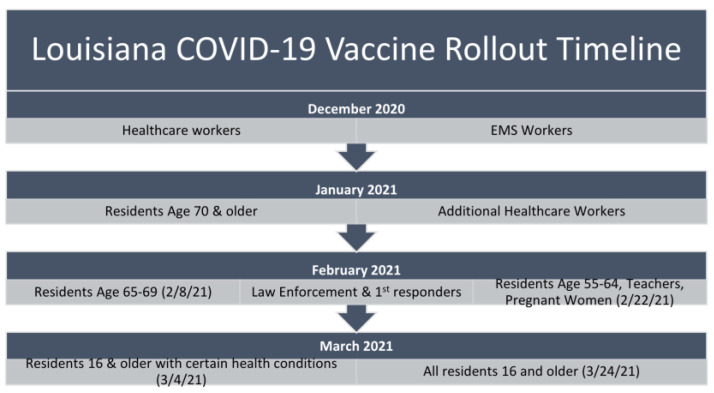
Louisiana COVID-19 Vaccine Phased Roll-out [23].

**Table 1 ijerph-19-16737-t001:** Baseline Demographics of participants based upon vaccine eligibility at time of survey (n = 487).

Variable	Not Eligible n = 212	Eligible n = 275	*p*-Value
Negative vaccine intent	86 (40.6%)	82 (29.8%)	0.009
Sex at birth (female)	126 (59.4%)	112 (40.9%)	0.0004
Age (mean, SD)	40 (15)	51 (18)	0.000
Education			
High school graduate, GED, or less	72 (34%)	99 (36%)	
Technical, Associates, Bachelor’s degree, or higher	139 (66%)	172 (64%)	0.395
Previous vaccine refusal (yes)	46 (21.7%)	65 (23.6%)	
Where do you usually go for medical care?			
Primary care doctor	156 (73.6%)	214 (77.8%)	---
Urgent care	17 (8%)	19 (6.9%)	0.68
Emergency room	8 (3.8%)	15 (5.5%)	0.417
Hospital	28 (13.2%)	18 (6.5%)	0.024
Essential worker (yes)	54 (25.5%)	85 (30.9%)	0.112
Insurance status (yes): private or public	172 (81.1%)	232 (84.4%)	0.206
COVID-19 experiences and impact			
Have you ever received a COVID-19 test	147 (69.3%)	180 (65.5%)	0.238
Ever tested positive for COVID-19	18 (12.2%)	38 (21.1%)	0.024
Family members have received the COVID-19 vaccine	116 (54.7%)	220 (80%)	0.000
COVID-19 negatively impacted income	92 (43.4%)	113 (41.1%)	0.338
Self-identified risk for getting COVID-19 (low or none)	104 (49.1%)	141 (50.9%)	0.347
Postponed medical care due to the pandemic	54 (25.5%)	86 (31.3%)	0.096
COVID-19 perception of response and accessibility (disagree or strongly disagree)			
I know where me or my family can receive vaccine	46 (22.6%)	36 (13.1%)	0.009
Government response to pandemic is adequate	112 (52.8%)	107 (38.9%)	0.001
Vaccine is easy for African Americans to get	63 (29.7%)	69 (25.1%)	0.15

COVID-19, coronavirus 2019; GED, graduate equivalency degree; SD, standard deviation.

**Table 2 ijerph-19-16737-t002:** Results of Bivariate Analysis of Intent to Receive Vaccine.

Variable	Unadjusted OR (95% CI)
Eligible for vaccine at time of survey	1.61 (1.1, 2.34)
Sex at birth (female)	1.53 (1.05, 2.23)
Age (mean, SD)	1.05 (1.04, 1.07)
Previous vaccine refusal (yes)	0.18 (0.12, 0.29)
Where have you sought medical care in the past two years?	
Primary care doctor	---
Urgent care	0.29 (0.14, 0.57)
Emergency room	0.21 (0.09, 0.52)
Hospital	0.57 (0.304, 1.07)
Essential worker (yes)	0.87 (0.58, 1.32)
Insurance status (yes): private or public	2.39 (1.49, 3.87)
COVID-19 experiences and impact	
Ever received a COVID-19 test	0.96 (0.903, 1.02)
Ever tested positive for COVID-19	0.78 (0.43, 1.43)
Family members have received the COVID-19 vaccine	2.8 (1.88, 4.18)
COVID-19 negatively impacted income	0.68 (0.47, 0.99)
Self-identified risk for getting COVID-19 (low or none)	0.56 (0.39, 0.83)
Postponed medical care due to the pandemic (Yes)	2.56 (1.61, 4.06)
COVID-19 perception of response and accessibility (disagree or strongly disagree)	
I know where me or my family members can sign-up to receive the vaccine	0.79 (0.48, 1.23)
The government vaccine roll-out has been fair	0.9 (0.58, 1.4)
Government response to pandemic is adequate	0.82 (0.56, 1.19)
Vaccine is easy for African Americans to get	1.07 (0.7, 1.64)

CI, confidence interval; COVID-19, coronavirus 2019; OR, odds ratio.

**Table 3 ijerph-19-16737-t003:** Results of Multivariable Analysis of Final Model of Positive Vaccine Intent.

	Unadjusted OR (95% CI)	Adjusted OR (95% CI)
Eligible for vaccine at time of survey	1.61 (1.1, 2.34) *	2.07 (1.28, 3.35) *
Sex at birth (female)	1.53 (1.05, 2.23) *	1.45 (0.88, 2.37)
Previous vaccine refusal (yes)	0.18 (0.12, 0.29) *	0.14 (0.08, 0.24) *
Where have you sought medical care in the past two years?		
Primary care doctor	---	---
Urgent care	0.29 (0.14, 0.57) *	0.43 (0.18, 1.03)
Emergency room	0.21 (0.09, 0.52) *	0.15 (0.05, 0.47) *
Hospital	0.57 (0.304, 1.07)	0.87 (0.39, 1.92)
Insurance status yes (private or public)	2.39 (1.49, 3.87) *	1.73 (0.92, 3.25)
COVID-19 experiences and impact		
COVID-19 negatively impacted income	0.68 (0.47, 0.99) *	0.79 (0.49, 1.28)
Self-identified risk for getting COVID-19 (low or none)	0.56 (0.39, 0.83) *	0.68 (0.42, 1.08)
Postponed medical care due to the pandemic	2.56 (1.61, 4.06) *	3.24 (1.81, 5.81) *

* *p*-value < 0.05, CI, confidence interval; OR, odds ratio.

## Data Availability

Not applicable.

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
