# Peer review of "Assessing the Impact of COVID-19 Phased Vaccine Eligibility on COVID-19 Vaccine Intent among African Americans in Southeastern Louisiana: A Community-Based, Cohort Study"

_ijerph, 2022, doi:10.3390/ijerph192416737_

Round 1

Reviewer 1 Report

The paper concerns an impact of the COVID-19 vaccine eligibility on a vaccination intent in the African Americans community of Southern Luisiana. The performed research was based on a prospective cohort study. The used statistical methods are rather standard but are properly chosen for this type of analysis. The Authors also properly identified the limitations of the study. Some of the obtained results are interesting. Among others, it follws from the analysis that persons who are classified as those ones with low priority for vaccination because they are not in high risk groups may think that vaccination is not necessary for them. In other words, it may lower their intent to being vaccinated. Although this observation may seem to be quite simple (or even almost obvious), it is important, since on one hand, the prioritization was/is necessary, but on the other hand, it leads to this negative effect. So, some ideas how to avoid this negative consequences would be very useful. Moreover, it would be good if the Authors could compare the observed situation in the studied community with similar observations for other communities known from the literature, if there are any.

Summarizing, the paper is interesting and well written and could be considered for a possible publication after taking into account the above mentioned suggestions.

Author Response

Thank you for your review. For the "minor improvements:

1) Background - We have added on additional article which was just recently published. This is a unique analysis without much information from the literature.

2) Methods - We have expanded on the methods section

"So, some ideas how to avoid this negative consequences would be very useful. Moreover, it would be good if the Authors could compare the observed situation in the studied community with similar observations for other communities known from the literature, if there are any."

We have included information on how to potentially avoid the impact of phased eligibility on vaccine intent in our revision. We have also found only one other study in Canada which we have referenced.

We hope our revisions will sufficiently address the concerns and we appreciate your feedback.

Reviewer 2 Report

IJERPH - 2043833

Thank you for asking me to review this manuscript. First, this is a well-prepared document. It is professionally written. I found a few edits related to grammar and spelling, see below for these. The formatting of the paper was clear, with ease of readability throughout.

The title seems appropriate and without duplication to other titles of similar articles. I found the information and analysis appropriate. The abstract, introduction, Intervention and implementation, place time and persons, purpose, evaluation, and public health significance were aligned and complete. The references and tables were appropriately constructed and utilized.

Comments/concerns and edit suggestions:

1.      Page 2, line 72. Paragraph beginning with ‘Using a community-based ….. among healthcare disparity communities…..  I recommend deleting disparity.

2.      Page 3, last paragraph on page. This has much information in this paragraph describing the population studied and locations selected. The authors have information included in this paragraph about a gap in the literature on college students, African American men under-represented, and the survey showing disproportionately female. I suggest two paragraphs: one describing the population and locations, and the other discussing the other information.

The authors pointed out the limitations related to how the survey participants were chosen, adding in more locations that may bring in a more well-rounded participant groups, which were important to note. The other limitation mentioned was that there may have been many who were not included the vaccine eligible groups that may have thought they did not need the vaccine because they were not in the first few eligible groups. This premise overshadows the whole study.

The conclusion was appropriate. This report and subsequent results are interesting. The authors acknowledged there is a gap in the literature about the non-eligible group(s) being less likely to receive the vaccine. They correctly ascertained the need for further study.

References were appropriate; citations were accurately used.

Title is a bit lengthy but accurately describes the study.

Again, thank you for allowing me to review this manuscript.

Author Response

Thank you for reviewing our article and we are incorporating the recommended revisions.

"Page 2, line 72. Paragraph beginning with ‘Using a community-based ….. among healthcare disparity communities…..  I recommend deleting disparity."

Disparity is a term that is commonly employed in the literature as an alternate reference for "marginalized" or "under represented." We will look at alternate terms.

"Page 3" - We split the paragraph as recommended.

Thank you for your feedback and time.  It is much appreciated.

Reviewer 3 Report

Al-Dahir et al. conducted a community-based study on COVID-19 vaccine intent among African Americans in southeastern Louisiana.  The finding in this study leads to an important public health gap that must be addressed: Among racial/ethnic minorities, perceptions surrounding vaccine eligibility and self-identified risk should be further explored to improve COVID-19 vaccination. The authors also discussed the limitation of this study in the discussion section. The English usage of this manuscript is clear and professional. However, the manuscript contains certain weaknesses: first, the sample size is too small; second, the study is based on self-reported vaccine completion and intent, not scientific data; third, some concerns about the statistical data should be addressed, including high p values. Please see my detailed comments below:

(1) The keywords selected in the keyword section are either too vague or not reflecting the topic of this manuscript well. Other keywords may be added to better attract readers and give them an overview of the manuscript. By the way, please fix the format of the keywords, so it goes like “Keywords: Vaccine Intent; Vaccine Hesitancy; COVID-19 vaccine”.

(2)  In Table 1, some of the P values are high; please provide an explanation or discussion in the main text.

(3) Line 127: why are there words in parentheses? This sentence doesn’t flow well. You might want to use a slash “/” instead of parentheses, such as “lesbian, gay, bisexual, transgender, queer/questioning, intersex, and asexual/agender”.  

Author Response

Dear Reviewer,

Thank you for your feedback and time.  Our replies are below:

1) Must be improved: We have expanded on the methods section to address this concern.

2) Can be improved - Research Design - we have incorporated verbiage around the importance of CBPR as a research design.  We also expanded on the length of the cohort follow-up to address 9-months as sufficient for vaccine eligibility to be expanded to the entire population.

3) Can be improved - Conclusions - We expanded on conclusions to address public health implications.

Sample Size - the sample size was calculated a priori based upon disparate vaccine uptake in the African-American community.  The sample size was exceeded from the need calculated and sufficient to determine significant differences.

Self reported vaccine completion - It is now clarified in the manuscript, but vaccine status was verified by presentation of a vaccine card during interview.

Table 1 - High p-values at baseline are important to indicate no difference at baseline based upon exposure.  We highlighted where differences existed at baseline (gender, education) and how this might impact the results. These differences were controlled for in the multivariate analysis.

Key words - we expanded the key words

We also made modifications where "/" were present.

Thank you for your feedback and we hope our revisions and replies address all the concerns.